## [Peer Review File · Communications Chemistry]

Reviewers' comments:

Reviewer #1 (Remarks to the Author):

Manuscript: All-polymer rechargeable fuel cell by Miyake et al.

The authors report the utilization of hydrogen storing polymer based on fluorenone/fluorenel for hydrogen storage in proton-exchange membrane fuel cells (PEMFCs). The authors compare the hydrogen evolution properties of fluorenone/fluorenel based hydrogen storable polymer (HSP) in Nafion and polyphenylene PEM (SPP-QP) based PEMFCs. The fluorenone/fluorenel for hydrogen storage is reported by Kato et al. (Nature Communications 7, Article number: 13032 (2016)) and found to be merely 0.29 wt%. Other than reporting the first use of HSPs in regenerative or rechargeable fuel cells, this manuscript does not represent significant chemical insights for HSPs.

The manuscript is generally well written, and text and figures are organized nicely. However, in my opinion, this manuscript may be suitable for more specific journal.

Here are my comments:

1. In abstract and title, authors have emphasized "All-polymer type" rechargeable fuel cells. Isn't all PEMFCs and regenerative fuel cells are "All-polymer type"? The authors should discuss the significance of the all-polymer system.
2. Title: All-polymer rechargeable fuel cell, appears generic and does not reflect the content of the manuscript.
3. The authors did not use the right control experiment, which makes comparison difficult. For example, there is no comparison with or without HSPs. As generated currents are low, therefore, there could be adsorbed Hydrogen responsible for these currents. Though fluorenone/fluorenel polymer properties are well explored recently by Kato et al., it should also be noted that it could only store 0.29 wt% Hydrogen. Moreover, Ir nanoparticles are reported to store Hydrogen (J. Am. Chem. Soc. 2012, 134, 16, 6893–6895).
4. Why was a different amount of HSPs used for SPP-QP and Nafion cells? It is a significant deviation, which further makes comparison difficult. As normalization per mg might not represent the actual amount of hydrogen evolution.
5. Provided that Nafion cell has ~150 wt% more HSP, it should compensate for the losses from hydrogen crossover, shouldn't it?

Reviewer #2 (Remarks to the Author):

The authors report the combination of a novel polymer storage for hydrogen with a recently reported hydrocarbon membrane for low hydrogen crossover. Even though this polymer storage might remain a niche application – since hydrogen for mobile application will most likely not be able to compete in terms of cost with the dramatic improvements currently happening in Li-Ion Battery technology – the manuscript is a novel contribution to the state of the art. The use of a hydrocarbon membrane due to the lower crossover makes sense and shows a possible future application for those new materials.

However, prior to publication, several inconsistencies in the manuscript need to be fixed: In particular the methods section is not yet acceptable in its current state and requires several corrections. Additionally, there are several aspects in the main text to be reviewed by authors before a possible acceptance of the publication.

Queries:

- 1) To my opinion, a weak point of the manuscript is the comparison "Nafion" vs. "SSP-QP": According to your method section, you compare a 25 μm hydrocarbon membrane vs. a 50 μm Nafion membrane, right? Why? You could have used either NR-211 (25 μm) or a 50 μm hydrocarbon membrane? According to the HFR data in Figure 4 d) the resistance of Nafion is twice as high as for the hydrocarbon – this could be simply an effect of thickness. Please provide at least an iR-free representation of figure 4a) to account for this difference. Better: include data of either NR-211 or a 50 μm hydrocarbon membrane.
- 2) Similar thing: Page 11, last sentence "since SPP-QP was also more proton-conductive.. the resistance was 48% lower". This is not (fully) true. You obviously use a 50% thicker membrane here as comparison. Please change the phrasing here.
- 3) Methods: What is the IEC of the hydrocarbon membrane? Even if you refer to previous work, please provide more information on the SPP-QP (as you do for Nafion)
- 4) Methods (in general, applies for the entire paragraph): Please be more specific! E.g. the preparation of the electrode slurry is in a state, that nobody could reproduce the results – key informations are missing, please add: (specific amounts of solvents, what ball mill with which ball diameter and rotation speed, etc..) What specific spray-coating device was used? (device name, nozzle type), What hotpress?
- 5) Methods: GDL and Gasket (which material??) have the same thickness? This would mean, that you do not compress your cell? How can you seal it then?
- 6) Number of gaskets: There are discrepancies between the methods' section ("6 GDLs, 6 gaskets") and Figure 2, where only 2-3 GDLs and Gaskets can be seen. Please explain, what you actually assembled.
- 7) Different thickness of the hydrogen storage for hydrocarbon and Nafion cell: Am I right, that you actually used differently thick storage polymers for hydrocarbon (1.2 mm, 50 mg) and Nafion (2.4 mm, 122.5 mm)? Since this information appears in the method section only and not in the manuscript, please comment. This has obviously severe impact on the amount of hydrogen stored!

Reviewer #3 (Remarks to the Author):

This is a very interesting manuscript describing an entirely novel integrated approach for hydrogen supply to a fuel cell. There is clearly a lot of research that still remains to be done since the amount of hydrogen produced from such a polymer is low, and the Ir catalyst degrades during the hydrogenation/dehydrogenation cycles, however this does not detract from the interest and potential impact of the results described.

I recommend publication but suggest that the authors consider the following minor points

1. "RFC" is the abbreviation generally used for reversible fuel cells, and I am concerned that the use of the same abbreviation here could lead to confusion. The authors might consider using a different name/abbreviation.
2. Figure 1 refers to "Chemical structure of the PEM (SPP-QP)" whereas experimental results both with this polymer membrane and with Nafion are described in the manuscript. This part of the caption should refer to "Chemical structure of SPP-QP".
3. The manuscript does not mention that Nafion was used until well into the results section. The introduction section should explicitly state that both Nafion and SPP-QP are used in the membrane electrode assemblies.
4. What is the loading of Ir catalyst in the experiments described.
5. Could the authors expand on the type of mobile applications where this rechargeable FC could be useful.
6. The need for multiple GDLs and gaskets at the anode side is unclear. Could the authors improve their explanation of this, and also why twice the thickness of HSP and twice the number of GDLs were used in the Nafion cell as is the SPP-QP cell.

Response to the Reviewer 1

We very much appreciate the helpful comments of the Reviewer and have made the requested modifications.

Manuscript: All-polymer rechargeable fuel cell by Miyake et al.

The authors report the utilization of hydrogen storing polymer based on fluorenone/fluorenol for hydrogen storage in proton-exchange membrane fuel cells (PEMFCs). The authors compare the hydrogen evolution properties of fluorenone/fluorenol based hydrogen storable polymer (HSP) in Nafion and polyphenylene PEM (SPP-QP) based PEMFCs. The fluorenone/fluorenol for hydrogen storage is reported by Kato et al. (Nature Communications 7, Article number: 13032 (2016)) and found to be merely 0.29 wt%. Other than reporting the first use of HSPs in regenerative or rechargeable fuel cells, this manuscript does not represent significant chemical insights for HSPs.

The manuscript is generally well written, and text and figures are organized nicely. However, in my opinion, this manuscript may be suitable for more specific journal.

Here are my comments:

1. In abstract and title, authors have emphasized “All-polymer type” rechargeable fuel cells. Isn’t all PEMFCs and regenerative fuel cells are “All-polymer type”? The authors should discuss the significance of the all-polymer system.

As the reviewer commented, PEMFCs are basically all-polymer type in terms of polymer electrolyte membranes. In the present case, we emphasize ‘all-polymer type’ since even hydrogen is stored in polymer (hydrogen storable polymer; HSP). To clarify this point, the sentences have been modified on pages 2 and 4 as follows.

(page 2) The RCFCs are cycleable, at least up to 50 cycles. The features of this novel RCFC system, including safety, ease of handling, and light weight, suggest applications in mobile, light-weight hydrogen-based energy devices.

(page 4) Herein, we propose an ‘all-polymer type’ rechargeable PEMFC system for the first time, by applying the HSP sheet as a hydrogen storage medium inside the cell, which neither requires pressurized hydrogen tank nor cumbersome metal hydrides.

2. Title: All-polymer rechargeable fuel cell, appears generic and does not reflect the content of the manuscript.

We, for the first time, applied hydrogen storable polymer inside the anode of a fuel cell and confirmed its cycleability. The concept of hydrogen storable polymer as a hydrogen source for PEMFCs has not been previously proposed. Therefore, we believe that ‘All-polymer

rechargeable fuel cell' represents well the content of this paper. The title of the manuscript has been modified as follows.

“All-polymer rechargeable fuel cell: Proof of concept study”

3. The authors did not use the right control experiment, which makes comparison difficult. For example, there is no comparison with or without HSPs. As generated currents are low, therefore, there could be adsorbed Hydrogen responsible for these currents. Though fluorenone/fluorenol polymer properties are well explored recently by Kato et al., it should also be noted that it could only store 0.29 wt% Hydrogen. Moreover, Ir nanoparticles are reported to store Hydrogen (J. Am. Chem. Soc. 2012, 134, 16, 6893–6895).

According to the Reviewer's suggestion, we have measured H₂ absorbability of the Ir catalyst at 25 °C (1 atm of H₂). The results suggested that the Ir catalyst absorbed up to 58 mol% H₂, corresponding to 4.7 wt% of the total stored H₂ in the HSP sheet. The following sentence has been added on page 5.

The Ir catalyst could also absorb up to 58mol% hydrogen at 1 atm of H₂, which was substantially lower (ca. 4.7 wt%) than that stored in HSP.

4. Why was a different amount of HSPs used for SPP-QP and Nafion cells? It is a significant deviation, which further makes comparison difficult. As normalization per mg might not represent the actual amount of hydrogen evolution.

5. Provided that Nafion cell has ~150 wt% more HSP, it should compensate for the losses from hydrogen crossover, shouldn't it?

According to the Reviewer's suggestion, we have carried out additional fuel cell test with Nafion NRE-211 membrane (25 μm thick) and 44.7 mg of HSP, where the thickness of the membrane and amount of HSP were the same as those in SPP-QP cell. Unfortunately, the operable time was rather short for both Nafion NRE-211 and -212 cells, even when double amount of HSP was used. The results support importance of gas impermeability of PEMs to mitigate hydrogen/oxygen cross-over through the membrane. The following discussion together with Supplementary Figure 2 has been added on page 10.

Prior to the detailed fuel cell evaluation with our SPP-QP membrane, preliminary experiment was conducted with a commercially available Nafion NRE-211 membrane (25 μm thick). The NRE-211 cell was operable only for ca. 14 s at a constant current density of 10 mA cm⁻² with 44.7 mg of HSP (Supplementary Fig. 2). To increase the operation time, the membrane was replaced with a Nafion NRE-212 (50 μm thick) and larger amount of HSP (122.5 mg) was used. The cell was operable for ca. 17 s but still rather short in spite of the thicker membrane and larger amount of hydrogen source. We speculated that use of SPP-QP as gas impermeable polyphenylene-based PEM must enhance the operable time. Comparison of fuel cell performance is made for Nafion-NRE-212 and SPP-QP cells hereafter.

Supplementary Figure 2 | Fuel cell performance at a constant current density of 10 mA cm⁻². The RCFC was monitored at 80 °C and 100% RH, with flowing O₂ at 20 mL min⁻¹. The membrane thickness and loading amount of HSP were 25 μm and 44.7 mg for Nafion NRE-211 cell and 50 μm and 122.5 mg for Nafion NRE-212 cell, respectively.

Response to the Reviewer 2

We very much appreciate the helpful comments of the Reviewer and have made the requested modifications.

The authors report the combination of a novel polymer storage for hydrogen with a recently reported hydrocarbon membrane for low hydrogen crossover.

Even though this polymer storage might remain a niche application – since hydrogen for mobile application will most likely not be able to compete in terms of cost with the dramatic improvements currently happening in Li-Ion Battery technology – the manuscript is a novel contribution to the state of the art. The use of a hydrocarbon membrane due to the lower crossover makes sense and shows a possible future application for those new materials.

However, prior to publication, several inconsistencies in the manuscript need to be fixed: In particular the methods section is not yet acceptable in its current state and requires several corrections. Additionally, there are several aspects in the main text to be reviewed by authors before a possible acceptance of the publication.

Queries:

1. To my opinion, a weak point of the manuscript is the comparison “Nafion” vs. “SSP-QP”: According to your method section, you compare a 25 μm hydrocarbon membrane vs. a 50 μm Nafion membrane, right? Why? You could have used either NR-211 (25 μm) or a 50 μm hydrocarbon membrane?

According to the HFR data in Figure 4 d) the resistance of Nafion is twice as high as for the hydrocarbon – this could be simply an effect of thickness. Please provide at least an iR -free representation of figure 4a) to account for this difference. Better: include data of either NR-211 or a 50 μm hydrocarbon membrane.

According to the Reviewer’s suggestion, we have carried out additional fuel cell test with Nafion NRE-211 membrane (25 μm thick) and 44.7 mg of HSP, where the thickness of the membrane and amount of HSP were the same as those in SPP-QP cell. Unfortunately, the operable time was rather short for both Nafion NRE-211 and -212 cells, even when double amount of HSP was used. The results support importance of gas impermeability of PEMs to mitigate hydrogen/oxygen cross-over through the membrane. The following discussion together with Supplementary Fig. 2 has been added on page 10.

Prior to the detailed fuel cell evaluation with our SPP-QP membrane, preliminary experiment was conducted with a commercially available Nafion NRE-211 membrane (25 μm thick). The NRE-211 cell was operable only for ca. 14 s at a constant current density of 10 mA cm^{-2} with 44.7 mg of HSP (Supplementary Fig. 2). To increase the operation time, the membrane was replaced with a Nafion NRE-212 (50 μm thick) and larger amount of HSP (122.5 mg) was used. The cell was operable for ca. 17 s but still rather short in spite of the thicker membrane and larger amount of hydrogen source. We speculated that use of SPP-QP as gas impermeable polyphenylene-based PEM must enhance the operable time.

Comparison of fuel cell performance is made for Nafion-NRE-212 and SPP-QP cells hereafter.

Supplementary Figure 2 | Fuel cell performance at a constant current density of 10 mA cm⁻². The RCFC was monitored at 80 °C and 100% RH, with flowing O₂ at 20 mL min⁻¹. The membrane thickness and loading amount of HSP were 25 μm and 44.7 mg for Nafion NRE-211 cell and 50 μm and 122.5 mg for Nafion NRE-212 cell, respectively.

According to the Reviewer's suggestion, an iR-free representation of Fig. 4a has been added in Supplementary Fig. 4.

Supplementary Figure 4 | IR free representation of Fig. 4a (period 6, Fig. 3). Cell voltage as a function of operation time, which is normalized by HSP weight. The fuel cells were operated at 80 °C and 100% RH, in which the flow rate of O₂ was 20 mL min⁻¹.

2. Similar thing: Page 11, last sentence “since SPP-QP was also more proton-conductive.. the resistance was 48% lower”. This is not (fully) true. You obviously use a 50% thicker membrane here as comparison. Please change the phrasing here.

According to the Reviewer’s suggestion, the statement on ohmic resistance has been modified as follows on page 12.

Since SPP-QP was thinner and more proton-conductive, the ohmic resistance of the SPP-QP cell was ca. 21 mΩ cm², i.e., ca. 48% lower than that of the Nafion NRE-212 cell (Fig. 4d).

3. Methods: What is the IEC of the hydrocarbon membrane? Even if you refer to previous work, please provide more information on the SPP-QP (as you do for Nafion)

According to the Reviewer’s suggestion, the IEC of the SPP-QP membrane has been added on page 18 as follows.

The SPP-QP membrane¹⁰ (25 μm thick, ion exchange capacity (IEC) = 2.14 mmol g⁻¹) and the HSP sheet¹⁹ (containing 12.6 wt% of the Ir catalyst) were prepared according to the literature.

4. Methods (in general, applies for the entire paragraph): Please be more specific! E.g. the preparation of the electrode slurry is in a state, that nobody could reproduce the results –

key informations are missing, please add: (specific amounts of solvents, what ball mill with which ball diameter and rotation speed, etc..) What specific spray-coating device was used? (device name, nozzle type), What hotpress?

5. Methods: GDL and Gasket (which material??) have the same thickness? This would mean, that you do not compress your cell? How can you seal it then?

6. Number of gaskets: There are discrepancies between the methods' section ("6 GDLs, 6 gaskets") and Figure 2, where only 2-3 GDLs and Gaskets can be seen. Please explain, what you actually assembled.

According to the Reviewer's suggestion, detailed information on the preparation of catalyst paste has been added on page 18 as follows. Furthermore, MEA fabrication procedure has been updated with detailed information on hot-pressing, GDL, gasket, and their numbers on page 19 as follows.

(page 18) **Catalyst paste preparation.** A catalyst paste was prepared by mixing a commercial Pt/CB catalyst (1 g, TEC10E50E, Tanaka Kikinzoku Kogyo K. K.), 5 wt% Nafion dispersion (7.52 g, IEC = 0.95-1.03 mmol g⁻¹, D-521, Du Pont), deionized water (4.19 g), and ethanol (8.21 g) by ball milling with zirconia balls (ϕ = 5 mm) using planetary ball mill (PULVERISETTE 6, FRITSCH) at 270 rpm for 30 min. The mass ratio of the Nafion binder to the carbon support (N/C) was adjusted to 0.70.

(page 19) **MEA fabrication.** The SPP-QP cell was prepared as follows. The catalyst-coated membrane (CCM) was prepared by spraying the catalyst paste on both sides of the SPP-QP membrane (25 μ m thick, IEC = 2.14 mmol g⁻¹) by means of the pulse-swirl-spray machine (Nordson, nozzle type : A7A for Swirl ver. 3 Dual syringe, type2). The CCM was dried at 60 °C for 12 h and hot-pressed (in-house hot press machine) at 140 °C and 10 kgf cm⁻² for 3 min. The geometric area and the Pt-loading amount of the catalyst layer (CL) were 4.41 cm² and 0.50 \pm 0.05 mg cm⁻², respectively. For the cathode side, a gas diffusion layer (GDL, 29BC, SGL Carbon Group Co., Ltd., 230 μ m thick) and a gasket (silicon rubber - polyethylene naphthalate - silicon rubber (Maxell Kureha Co., Ltd), 200 μ m thick, quadratic prism (ca. 6 \times 6 cm²) hollowed out (ca. 2.1 \times 2.1 cm²) in the center), and for the anode side, a HSP sheet (50 mg, square frustum [1.5 mm thick, upper base ca. 1 \times 1 cm², lower base ca. 2 \times 2 cm²]), six GDLs (the same GDL as the above, but hollowed out (ca. 2 \times 2 cm²) in the center, 1.38 mm thick in total), and six gaskets (the same gasket as above, 1.2 mm thick in total) were placed on the CCM and mounted into a cell that had serpentine flow channels on both the anode and the cathode carbon separators. The Nafion NRE-212 cell using Nafion NRE-212 membrane (50 μ m thick, IEC = 0.98 mmol g⁻¹, Du Pont) was prepared in a similar manner; an HSP sheet (122.5 mg, square frustum [3.3 mm thick, upper base ca. 1 \times 1 cm², lower base ca. 2 \times 2 cm²]), twelve GDLs (the same GDL as the above, but hollowed out (ca. 2 \times 2 cm²) in the center, 2.76 mm thick in total), and twelve gaskets (the same gasket as above, 2.4 mm thick in total) were used for the anode side.

7. Different thickness of the hydrogen storage for hydrocarbon and Nafion cell: Am I right, that you actually used differently thick storage polymers for hydrocarbon (1.2 mm, 50 mg) and Nafion (2.4 mm, 122.5 mm)? Since this information appears in the method section only and not in the manuscript, please comment. This has obviously severe impact on the amount of hydrogen stored!

We agree with the Reviewer's opinion. We have carried out additional fuel cell test with Nafion NRE-211 membrane (25 μm thick) and 44.7 mg of HSP, where the thickness of the membrane and amount of HSP were the same as those in SPP-QP cell. Unfortunately, the operable time was rather short for both Nafion NRE-211 and -212 cells, even when double amount of HSP was used. The results support importance of gas impermeability of PEMs to mitigate hydrogen/oxygen cross-over through the membrane. The following discussion together with Supplementary Fig. 2 has been added on page 10.

Prior to the detailed fuel cell evaluation with our SPP-QP membrane, preliminary experiment was conducted with a commercially available Nafion NRE-211 membrane (25 μm thick). The NRE-211 cell was operable only for ca. 14 s at a constant current density of 10 mA cm^{-2} with 44.7 mg of HSP (Supplementary Fig. 2). To increase the operation time, the membrane was replaced with a Nafion NRE-212 (50 μm thick) and larger amount of HSP (122.5 mg) was used. The cell was operable for ca. 17 s but still rather short in spite of the thicker membrane and larger amount of hydrogen source. We speculated that use of SPP-QP as gas impermeable polyphenylene-based PEM must enhance the operable time. Comparison of fuel cell performance is made for Nafion-NRE-212 and SPP-QP cells hereafter.

Supplementary Figure 2 | Fuel cell performance at a constant current density of 10 mA cm^{-2} . The RCFC was monitored at 80 $^{\circ}\text{C}$ and 100% RH, with flowing O_2 at 20 mL min^{-1} .

The membrane thickness and loading amount of HSP were 25 μm and 44.7 mg for Nafion NRE-211 cell and 50 μm and 122.5 mg for Nafion NRE-212 cell, respectively.

Response to the Reviewer 3

We very much appreciate the helpful comments of the Reviewer and have made the requested modifications.

This is a very interesting manuscript describing an entirely novel integrated approach for hydrogen supply to a fuel cell. There is clearly a lot of research that still remains to be done since the amount of hydrogen produced from such a polymer is low, and the Ir catalyst degrades during the hydrogenation/dehydrogenation cycles, however this does not detract from the interest and potential impact of the results described.

I recommend publication but suggest that the authors consider the following minor points

1. "RFC" is the abbreviation generally used for reversible fuel cells, and I am concerned that the use of the same abbreviation here could lead to confusion. The authors might consider using a different name/abbreviation.

According to the Reviewer's suggestion, the abbreviation of our rechargeable fuel cell has been modified to "RCFC" throughout the paper.

2. Figure 1 refers to "Chemical structure of the PEM (SPP-QP)" whereas experimental results both with this polymer membrane and with Nafion are described in the manuscript. This part of the caption should refer to "Chemical structure of SPP-QP".

According to the Reviewer's suggestion, the caption of Figure 1 has been modified as follows. Furthermore, Figure 1 has been modified for better recognition.

Fig. 1 Conceptual diagram of the RCFC. An HSP sheet, which can release/fix hydrogen repeatedly, was attached onto the catalyst layer of the anode side. SPP-QP (or Nafion) membrane was used as PEM.

3. The manuscript does not mention that Nafion was used until well into the results section. The introduction section should explicitly state that both Nafion and SPP-QP are used in the membrane electrode assemblies.

According to the Reviewer's suggestion, the statement has included both Nafion and SPP-QP on page 5 as follows.

Fig. 1 further shows the chemical structure of the PEM (SPP-QP) used in this study¹⁰. The SPP-QP, which we have recently developed, is a fluorine-free, fully-aromatic-type PEM, whose gas barrier properties are far superior to that of a commercially-available, perfluorinated-type PEM such as Nafion. Hydrogen and oxygen gas permeabilities of SPP-QP (IEC of 2.4 mmol g⁻¹) at 80 °C and 90% relative humidity (RH) were 1.46×10^{-9} and 4.72×10^{-10} cm³ (STD) cm cm⁻² s⁻¹ cmHg⁻¹, respectively, compared to those (7.35×10^{-9} and 3.15×10^{-9} cm³ (STD) cm cm⁻² s⁻¹ cmHg⁻¹) of a Nafion NRE212 membrane. In addition, the SPP-QP membrane fulfills other required properties for fuel cell applications in terms of proton conductivity and stability (e.g., thermal/mechanical/chemical). RCFC performance is compared between SPP-QP and Nafion NRE-212 cells in details.

4. What is the loading of Ir catalyst in the experiments described.

According to the Reviewer's suggestion, information on the loading of Ir catalyst has been added on page 18 as follows.

The SPP-QP membrane¹⁰ (25 μm thick, ion exchange capacity (IEC) = 2.14 mmol g⁻¹) and the HSP sheet¹⁹ (containing 12.6 wt% of the Ir catalyst) were prepared according to the literature.

5. Could the authors expand on the type of mobile applications where this rechargeable FC could be useful.

We believe our RCFC may expand mobile applications where the current LIBs are not available in terms of the safety issues. The related sentence has been added on page 2 as follows.

The features of this novel RCFC system, including safety, ease of handling, and light weight, suggest applications in mobile, light-weight hydrogen-based energy devices.

6. The need for multiple GDLs and gaskets at the anode side is unclear. Could the authors improve their explanation of this, and also why twice the thickness of HSP and twice the number of GDLs were used in the Nafion cell as is the SPP-QP cell.

Since HSP was much thicker (1.5 mm thick for 50 mg of HSP and 3.3 mm for 122.5 mg of HSP, respectively) than GDL and gasket, multiple GDLs and gaskets were used to ensure tight seal. Figure 2 has been modified. The following sentence has been added on page 6.

To adjust the thickness with the HSP sheet (note that HSP was 1.5-3.3 mm thick), multiple GDLs and gaskets were used to ensure tight seal.

According to the Reviewer's suggestion, we have carried out additional fuel cell test with Nafion NRE-211 membrane (25 μm thick) and 44.7 mg of HSP, where the thickness of the membrane and amount of HSP were the same as those in SPP-QP cell. Unfortunately, the operable time was rather short for both Nafion NRE-211 and -212 cells, even when double amount of HSP was used. The results support importance of gas impermeability of PEMs to mitigate hydrogen/oxygen cross-over through the membrane. The following discussion together with Supplementary Fig. 2 has been added on page 10.

Prior to the detailed fuel cell evaluation with our SPP-QP membrane, preliminary experiment was conducted with a commercially available Nafion NRE-211 membrane (25 μm thick). The NRE-211 cell was operable only for ca. 14 s at a constant current density of 10 mA cm^{-2} with 44.7 mg of HSP (Supplementary Fig. 2). To increase the operation time, the membrane was replaced with a Nafion NRE-212 (50 μm thick) and larger amount of HSP (122.5 mg) was used. The cell was operable for ca. 17 s but still rather short in spite of the thicker membrane and larger amount of hydrogen source. We speculated that use of SPP-QP as gas impermeable polyphenylene-based PEM must enhance the operable time. Comparison of fuel cell performance is made for Nafion-NRE-212 and SPP-QP cells hereafter.

Supplementary Figure 2 | Fuel cell performance at a constant current density of 10 mA cm⁻². The RCFC was monitored at 80 °C and 100% RH, with flowing O₂ at 20 mL min⁻¹. The membrane thickness and loading amount of HSP were 25 μm and 44.7 mg for Nafion NRE-211 cell and 50 μm and 122.5 mg for Nafion NRE-212 cell, respectively.

REVIEWERS' COMMENTS:

Reviewer #1 (Remarks to the Author):

Manuscript: All-polymer rechargeable fuel cell: Proof of concept study by Miyake et al.,
I appreciate authors hard work to answer queries in the revised manuscript.

1. The OCV difference of 0.81 V (Nafion-212) vs 0.83 V (SPP-QP) is rather small. If possible, provide hydrogen permeability values for both Nafion and SPP-QP membranes.
2. Provide some basic details about HSPs such as polymer type, redox reactions to store hydrogen etc.

Reviewer #2 (Remarks to the Author):

The authors added the required information. In particular the method section was significantly improved and enables now a reproduction of the results. Therefore I agree with publishing the article in its current form.

Reviewer #3 (Remarks to the Author):

The authors have made the suggested revisions to their manuscript and have performed additional experiments. Although much work remains to be done on this system, it is a novel and interesting concept and I recommend publication of the manuscript.

Response to the Reviewer 1

We very much appreciate the helpful comments of the Reviewer and have made the requested modifications.

Manuscript: All-polymer rechargeable fuel cell: Proof of concept study by Miyake et al., I appreciate authors hard work to answer queries in the revised manuscript.

1. The OCV difference of 0.81 V (Nafion-212) vs 0.83 V (SPP-QP) is rather small. If possible, provide hydrogen permeability values for both Nafion and SPP-QP membranes.

The hydrogen and oxygen permeabilities of the membranes are provided on page 5 as follows.

Hydrogen and oxygen gas permeabilities of SPP-QP (IEC of 2.4 mmol g⁻¹) at 80 °C and 90% relative humidity (RH) were 1.46×10^{-9} and 4.72×10^{-10} cm³ (STD) cm cm⁻² s⁻¹ cmHg⁻¹, respectively, compared to those (7.35×10^{-9} and 3.15×10^{-9} cm³ (STD) cm cm⁻² s⁻¹ cmHg⁻¹) of a Nafion NRE212 membrane.

2. Provide some basic details about HSPs such as polymer type, redox reactions to store hydrogen etc.

According to the reviewer's suggestion, the polymer type of HSP is described on page 4 as follows.

The amorphous, non-conjugated HSP can be molded into a bendable, safe, and lightweight sheet form,...

Furthermore, Figure 1 has been modified to include redox reaction of HSP with hydrogen fixation/release.

Response to the Reviewer 2

The authors added the required information. In particular the method section was significantly improved and enables now a reproduction of the results. Therefore I agree with publishing the article in its current form

We very much appreciate your positive comments on the revised manuscript.

Response to the Reviewer 3

The authors have made the suggested revisions to their manuscript and have performed additional experiments. Although much work remains to be done on this system, it is a novel and interesting concept and I recommend publication of the manuscript.

We very much appreciate your positive comments on the revised manuscript.